# High Efficiency In Vitro Wound Healing of *Dictyophora indusiata* Extracts via Anti-Inflammatory and Collagen Stimulating (MMP-2 Inhibition) Mechanisms

**DOI:** 10.3390/jof7121100

**Published:** 2021-12-20

**Authors:** Yasir Nazir, Pichchapa Linsaenkart, Chiranan Khantham, Tanakarn Chaitep, Pensak Jantrawut, Chuda Chittasupho, Pornchai Rachtanapun, Kittisak Jantanasakulwong, Yuthana Phimolsiripol, Sarana Rose Sommano, Jiraporn Tocharus, Salin Mingmalairak, Anchali Wongsa, Chaiwat Arjin, Korawan Sringarm, Houda Berrada, Francisco J. Barba, Warintorn Ruksiriwanich

**Affiliations:** 1Department of Pharmaceutical Sciences, Faculty of Pharmacy, Chiang Mai University, Chiang Mai 50200, Thailand; ynchem@yahoo.com (Y.N.); pichchapa_li@cmu.ac.th (P.L.); ckhantham@gmail.com (C.K.); tanakarn_c@cmu.ac.th (T.C.); pensak.j@cmu.ac.th (P.J.); chuda.c@cmu.ac.th (C.C.); 2Cluster of Research and Development of Pharmaceutical and Natural Products Innovation for Human or Animal, Chiang Mai University, Chiang Mai 50200, Thailand; sarana.s@cmu.ac.th (S.R.S.); korawan.s@cmu.ac.th (K.S.); 3Cluster of Agro Bio-Circular-Green Industry, Faculty of Agro-Industry, Chiang Mai University, Chiang Mai 50100, Thailand; pornchai.r@cmu.ac.th (P.R.); jantanasakulwong.k@gmail.com (K.J.); yuthana.p@cmu.ac.th (Y.P.); 4Faculty of Agro-Industry, Chiang Mai University, Chiang Mai 50100, Thailand; 5Faculty of Medicine, Chiang Mai University, Chiang Mai 50200, Thailand; jiraporn.tocharus@cmu.ac.th (J.T.); mingmalairak_s@yahoo.com (S.M.); 6Department of Animal and Aquatic Sciences, Faculty of Agriculture, Chiang Mai University, Chiang Mai 50200, Thailand; awongsa@hotmail.com (A.W.); chaiwat_arjin@cmu.ac.th (C.A.); 7Department of Preventive Medicine and Public Health, Food Science, Toxicology and Forensic Medicine, Faculty of Pharmacy, Universitat de València, 46100 Valencia, Spain; houda.berrada@uv.es (H.B.); francisco.barba@uv.es (F.J.B.)

**Keywords:** anti-inflammatory, bamboo mushroom, collagen stimulating activity, *Dictyophora indusiata*, matrix metalloproteinase-2 activity, wound healing

## Abstract

*Dictyophora indusiata* or *Phallus indusiatus* is widely used as not only traditional medicine, functional foods, but also, skin care agents. Biological activities of the fruiting body from *D. indusiata* were widely reported, while the studies on the application of immature bamboo mushroom extracts were limited especially in the wound healing effect. Wound healing process composed of 4 stages including hemostasis, inflammation, proliferation, and remodelling. This study divided the egg stage of bamboo mushroom into 3 parts: peel and green mixture (PGW), core (CW), and whole mushroom (WW). Then, aqueous extracts were investigated for their nucleotide sequencing, biological compound contents, and wound healing effect. The anti-inflammatory determination via the levels of cytokine releasing from macrophages, and the collagen stimulation activity on fibroblasts by matrix metalloproteinase-2 (MMP-2) inhibitory activity were determined to serve for the wound healing process promotion in the stage 2–4 (wound inflammation, proliferation, and remodelling of the skin). All *D. indusiata* extracts showed good antioxidant potential, significantly anti-inflammatory activity in the decreasing of the nitric oxide (NO), interleukin-1 (IL-1), interleukin-1 (IL-6), and tumour necrosis factor-α (TNF-α) secretion from macrophage cells (*p* < 0.05), and the effective collagen stimulation via MMP-2 inhibition. In particular, CW extract containing high content of catechin (68.761 ± 0.010 mg/g extract) which could significantly suppress NO secretion (0.06 ± 0.02 µmol/L) better than the standard anti-inflammatory drug diclofenac (0.12 ± 0.02 µmol/L) and their MMP-2 inhibition (41.33 ± 9.44%) was comparable to L-ascorbic acid (50.65 ± 2.53%). These findings support that CW of *D. indusiata* could be an essential natural active ingredient for skin wound healing pharmaceutical products.

## 1. Introduction

Mushroom is rich in nutrients and bioactive compounds. Edible mushroom is regularly being used as food, dietary supplements as well as cosmeceutical products such as anti-ageing, moisturizing, and skin lightening. Since, mushroom polysaccharides and polyphenol contents were involved in antioxidant and immunomodulatory activities [1,2]. *Dictyophora indusiata* or *Phallus indusiatus* or bamboo mushroom is a fungus belonging to the family *Phallaceae*. The bioactive compositions in immature and mature stages of this mushroom are different. The egg or immature mushroom produces viscous mucilage and high phenolic contents [3,4]. The mature stage of this mushroom contains many bioactive compounds, for example, polysaccharides, amino acids, terpenoids, and alkaloids [5,6]. Many studies revealed that the fruiting body of *D. indusiata* showed various impressive activities such as anti-obesity [7], and neuroprotective effect for treating Alzheimer’s disease [8]. Furthermore, *D. indusiata* also has considerable bioactivities with cosmeceutical potentials such as antioxidant, anti-tyrosinase, antimicrobial properties [9,10,11]. Interestingly, most research papers reported that polysaccharides from the fruiting body of *D. indusiata* contributed to immune modulation [12,13,14] and considered as a nutraceutical supplement in Chinese remedy. The superior anti-inflammatory activity of *D. indusiata* can be applied both via the oral and topical routes. Since the skin inflammation is a process that happens when the skin is damaged and wounded.

The skin wound repairing process included 4 main approaches, hemostasis, inflammation, proliferation, and tissue remodelling. Firstly, fibrin formation is the crucial step in coagulation cascades to stop blood loss [15]. During this phase, the number of soluble mediators such as platelet-derived growth factor (PDGF), insulin-like growth factor-1 (IGF-1), epidermal growth factor (EGF), fibroblast growth factor (FGF), transforming growth factor-β (TGF-β), and vascular endothelial cell growth factor (VEGF) are released from platelets [16]. Secondly, inflammation phase, macrophages migrate and defend microbes, attract other macrophage cells, and secrete cytokines and protease (elastase and collagenase) which degrade the skin extracellular matrix (ECM) components [17]. Inflammatory cytokines containing interleukin-1 (IL-1) and tumour necrosis factor-α (TNF-α) activated proteases production and apoptosis in fibroblasts. Thirdly, the promotion of cell proliferation and restoration of the matrix, fibroblasts, endothelial cells, and keratinocytes in tissue remodelling are conducted. Growth factors and cytokines are synthesized to promote cell proliferation, develop new capillary formation, and produce new ECM components [18,19]. Subsequently, matrix metalloproteinases (MMPs) will remove injured matrix proteins and then fibroblasts release lysyl oxidase to link collagen at ECM in the scar forming.

However, the most common wound-healing problem is the hypertrophic scar or keloid resulting from prolonged inflammation [20] and the action of matrix metalloproteinases enzymes [15]. Thus, the suppression of the inflammatory cytokines improves skin structure and barrier functions in the skin wounds [21] and MMP-2 inhibitory potential could restore the balance of collagen production [22] resulting in the successful wound healing process without the hypertrophic scar.

Moreover, most research papers reported various bioactivities of the fruiting body from *D. indusiata*, while the studies on the application of immature bamboo mushroom extracts as cosmeceutical or pharmaceutical applications were only a few. Therefore, this study aimed to investigate total phenolic, flavonoid, and polysaccharide contents, then further determine the antioxidant capabilities of aqueous extracts from 3 parts of *D. indusiata* mushroom. In addition, these extracts were assessed for wound healing activity using RAW 264.7 macrophages and *hTRT* fibroblasts which compared to the standard substances. The results from this study may consider as bioactive sources for pharmaceutical and/or cosmeceutical applications.

## 2. Results and Discussion

### 2.1. Identification of Dictyophora indusiata

#### 2.1.1. Specific Primers Designing and Selection

The primers: 18S ribosomal DNA (rDNA)-internal transcribed spacer 1 (ITS 1)-5.8S rDNA-internal transcribed spacer 2 (ITS 2)-28S rDNA and ITS1-5.8S rDNA-ITS2 sequences were obtained from *D. indusiata* f. lutea [23] and *D. indusiata* strain ASI 32001 [24], respectively. The designed specific primers for the identification of the sample are illustrated in Table 1. The PCR products of the sample gave one band with the expected size of 107 bp (Dict 01 primer), 150 bp (Dict 03 primer), 650 bp (Dict 04 primer), and 100 bp (Dict 05 primer), respectively. Although, Dict 02 and Dict 06 primers could not amplify DNA from the sample. The gel purified PCR products are represented in Figure 1. Therefore, the specific primers: Dict 03 and Dict 04 were chosen to identify the sample further.

#### 2.1.2. Nucleotide Sequences and Phylogenetic Relationships

The complete nucleotide sequences resulting from Dict 03 primer and Dict 04 primer are illustrated in Table 2. The sequences of the sample were blasted compared to the sequences of the available fungi species from GenBank. The nucleotide sequences resulting from Dict 03 primer matched 98% *Dictyophora indusiata* strain ZS03 (accession number MH464257.1), *Dictyophora echinovolvata* strain ASI 32010 (accession number AF324167.2), and *Dictyophora echinovolvata* strain ASI 32002 (accession number AF324164.2). The sequences of the sample also belonged to *Phallus echinovolvatus* voucher GDGM 79013 (accession number MN613536.1), *Phallus echinovolvatus* strain GDGM 79020 (accession number MN523216.1), *Dictyophora echinovolvata* strain ASI 32014 (accession number AF324168.2), *Dictyophora echinovolvata* strain ASI 32007 (accession number AF324165.2), and *Dictyophora echinovolvata* strain ASI 32008 (accession number AF324166.2) (97% identity) which obtained from Dict 04 primer. Furthermore, the percentages of nucleotide identity between the sample in the present study and the fungi species obtained from GenBank are displayed in Table 3 and Table 4. The phylogenetic trees were assessed using the genetic distances from neighbor joining (NJ) method with 1000 bootstrap replications. The trees are presented in Figure 2 and Figure 3.

### 2.2. Bioactive Compounds and Antioxidant Activities

The *D. indusiata* extracts were prepared from the immature stage of mushroom after water extraction. The extraction yields of PGW, CW and WW were 21.11%, 51.48% and 10.15% of dry weight, respectively. The amount of bioactive compound from the 3 types of egg *D. indusiata* extracts is illustrated in Table 5. The major component from immature bamboo mushroom was phenolic compound followed by polysaccharide and flavonoid, respectively. The polyphenol compositions were presented in Table 6. Catechin (68.761 ± 0.010 mg/g extract) was a dominant compound in the CW, followed by *p*-coumaric acid (7.931 ± 0.939 mg/g extract). Nevertheless, the major polyphenol component of PGW and WW was *p*-coumaric acid. In contrast, the previous study reported that the β-D-glucan polysaccharide backbone, monosaccharide compositions of glucose, mannose, and galactose were found in the aqueous extract of *D. indusiata* fruiting body that prepared by water extraction [5].

In the current study, PGW showed not only the highest contents of phenolic (2.55 ± 0.36 mg GAE/g extract), flavonoid (0.05 ± 0.02 mg CE/g extract), and polysaccharide (2.22 ± 0.29 µg GE/g extract) but also the highest FRAP value as 6.18 ± 0.08 µM Fe^2+^/g extract. However, the PGW also has lower activity than medical plant extract for DPPH, ABTS, and FRAP [25,26]. Moreover, CW presented greater antioxidant potency (DPPH assay, ABTS assay and metal chelation) than PGW, which may be due to a high catechin content, as the results of their antioxidant activities are summarized in Table 7.

DPPH radical scavenging assay measured antioxidant capacity from reducing the deep violet colour of the stable DPPH radicals to the yellow colour of nonradical DPPH [27]. The hydrogen-donating compounds such as ferulic acid, γ-oryzanol, unsaturated fatty acid, polysaccharide could scavenge the DPPH radical to nonradical form [28,29]. For ABTS^+^ assay, the absorbance of ABTS^+^ could decrease by losing their nitrogen atom [27]. The hydroxy groups found in phenolic contents are able to scavenge DPPH and ABTS radicals [30]. Furthermore, ketonic oxygen, degree of methoxylation, the amount of methyl and methylene within the structure of phenolic compounds indicated the antioxidant potential using DPPH and ABTS assay [31,32]. Catechin consists of numerous hydroxy groups which gave antioxidant effects in vitro and in vivo [33,34]. Furthermore, *p*-coumaric acid is a hydroxyl derivative of cinnamic acid that can directly scavenge the reactive oxygen species (ROS) [35,36]. These phenolic compounds may contribute to the synergistic effect on the antioxidant test. Moreover, FRAP assay was evaluated by the reducing ferrous ion (Fe^2+^) from ferric ion (Fe^3+^) [27]. The metal chelating assay was performed by measurement of ferrozine–Fe^2+^ complex formation. Good metal chelator could have a better competitive binding with ferrous ion than ferrozine in the system [37].

In previous literature, *D. indusiata* found dictyophorines A and B as well as dictyoquinazol A, B, and C, which contained ketonic oxygens and hydroxy groups [38]. A previous study also reported that immature and mature *D. indusiata* mushroom comprised high phenolic compounds and contributed to their antioxidant potential [3]. These bioactive substances may affect the scavenging and chelating capacities of the extracts. Free radicals could injure cells and tissues such as membrane lipids, enzymes, and nucleic acids, leading to destructive wound repairing [39,40]. Thus, the antioxidant activities of *D. indusiata* extracts might prevent cells and tissues damage resulting in the alleviation of skin wounds.

### 2.3. Anti-Inflammatory Activity of Dictyophora indusiata Aqueous Extracts

#### 2.3.1. Non-Cytotoxic Concentration by the Sulforhodamine B (SRB) Assay on Macrophages

The non-cytotoxic concentration of *D. indusiata* aqueous extracts (PGW, CW, and WW), as evaluated by the SRB assay, was 20 µg/mL. This concentration, which provided more than 80% cell viability [19], was selected for further analyses.

#### 2.3.2. Anti-Inflammatory Activity

The inhibitions of NO, IL-1, IL-6 and TNF-α production from 3 types of *D. indusiata* extracts (PGW, CW, and WW) in RAW 264.7 cells, are displayed in Figure 4. For the measurement of nitrite concentration, all extracts significantly decreased nitrite secretion in LPS-induced RAW 264.7 macrophages in a dose-dependent and time-dependent manner (*p* < 0.05). Interestingly, the nitrite concentration of CW at 20 µg/mL (0.06 ± 0.02 µmol/L) was significantly lower than of diclofenac sodium (0.12 ± 0.02 µmol/L) after 72 h of incubation (*p* < 0.05). The assessment of IL-1 secretion, the results in IL-1 expression of PGW, CW, and WW (at 20 and 0.2 µg/mL) were significantly de-escalated in LPS-treated RAW 264.7 cells after 24, 48, and 72 h (*p* < 0.05). For IL-6 production, at the concentration of 20 µg/mL, PGW, CW, and WW were profoundly inhibited IL-6 secretion comparable with the diclofenac treatment group at 24, 48, and 72 h tested time. The evaluation of TNF-α secretion, 3 types of extracts at the 20 and 0.2 µg/mL concentration illustrated significant in lowering TNF-α production in LPS-stimulated macrophages after 24, 48, and 72 h (*p* < 0.05). Especially, 20 µg/mL of PGW and CW depicted complete inhibition of TNF-α secretion at 24 h of incubation.

This result agreed with a previous study, showing that crude polysaccharide from *D. indusiata* extracts could reduce inflammatory cytokines (IL-1β, IL-6, and TNF-α) in antibiotic-induced intestinal dysbiosis in mice [41]. Furthermore, ethanolic extracts of *D. indusiata* could diminish cytokines production in RAW 264.7 cells [42]. Nevertheless, in earlier research, polysaccharide from this mushroom escalated NO, IL-1β, IL-6, and TNF-α production [13], activated IL-1β, and IL-18 secretion in LPS-induced macrophages [14], and IL-1β, IL-6, and IL-18 were enhanced in colitis mice [43]. Recently published studies reported that the bioactive compounds from various sources could regulate the immune system [39,44,45,46,47,48].

IL-1β, IL-6, and TNF-α were potent cytokines that dominated in skin wounds [49]. The previous study reported that IL-1, IL-6, and IL-8 significantly upregulated in wound fluids from patients with chronic wounds [50]. Furthermore, non-healing wounds of patients with comorbidity (diabetes and/or cardiovascular disease) found the increasing in cytokines (IL-1β, IL-4, IL-6, IL-8, TNF-α) and growth factors (FGF2, monocyte chemoattractant protein-1 (MCP-1), macrophage inflammatory protein-1α (MIP-1α), VEGF-A, and PDGF-BB) levels [51]. So, excessive secretion of cytokines would interfere normal healing process. Moreover, these inflammatory cytokines also caused cutaneous inflammation such as hidradenitis suppurativa [52,53]. MMP-1 and TNF-α contributed to comedones and papules in acne lesions [54]. In addition, *Propionibacterium acnes* that colonized in the pilosebaceous unit of acne skin can induce the expression of protease activated receptors (PARs), TLRs, TNF-α, IL-8, IL12, IL-1, MMPs, interferon-γ (IFN-γ), and granulocyte-macrophage colony-stimulating factor (GM-CSF) [55,56]. Although, IL-6 took part in the inflammatory phase of skin lesion, but IL-6 can stimulate fibroblast migration acceleration during the early stage of wound healing [57,58].

Consequently, the anti-inflammatory potential of CW could soothe chronic skin inflammation and wound healing impairment. Similar to *Brassica oleracea* extract which could suppress not only the levels of TNF-α, IFN-γ, IL-6, MCP-1, but also swelling and erythema in dermatitis induced-mice [59]. This corresponds with *Momordica charantia* extract can inhibit nuclear factor-κB (NF-κB) and mitogen-activated protein kinases (MAPK) activation leading to decrease the cytokine (IL-8, IL-1β, and TNF-α) production in *P. acnes*-stimulated monocytic cells [60].

### 2.4. Wound Healing Activity of Dictyophora indusiata Aqueous Extracts

#### 2.4.1. Non-Cytotoxic Concentration by the Sulforhodamine B (SRB) Assay on Fibroblasts

The non-cytotoxic concentration of bamboo mushroom aqueous extracts (PGW, CW, and WW), as determined by the SRB assay, was 1 mg/mL This concentration, which provided more than 80% cell viability [19], was selected for MMP-2 inhibitory analysis.

#### 2.4.2. Gelatinolytic Activity (Zymography) of MMP-2 Inhibition

The *D. indusiata* aqueous extracts (PGW, CW, and WW) and L-ascorbic acid standard showed the inhibitory effects of the gelatinolytic activity on MMP-2 expression, are illustrated in Figure 5. PGW presented the highest MMP-2 inhibition of 59.63 ± 8.31%, followed by CW (41.33 ± 9.44%) and WW (16.33 ± 2.91%), respectively. The L-ascorbic acid illustrated the inhibitory effect of 50.65 ± 2.53% while concanavalin A exhibited stimulation of gelatinolytic activity as −52.46 ± 2.53%. Interestingly, MMP-2 inhibitory abilities of PGW and CW indicated no significant difference to L-ascorbic acid. Our previous study of bamboo mushroom ethanolic extracts was also reported their MMP-2 inhibitory activity [42].

Pro-MMP-2, MMP-2, MMP-8, and MMP-9 were persistently elevated in wounds from diabetic patients and caused an imbalance of MMP and TIMP-2, leading to healing failure [61]. Additionally, higher MMP-2 mRNA levels in the wound corresponded with IFN-γ defection [62]. Furthermore, the excessive production of MMPs is associated with scar formation. The previous literature showed that the levels of type 1 collagen, MMP-1, MMP-2, and TIMP-1 in keloid fibroblasts was produced higher than those of normal dermal fibroblasts [63]. Similarly, MMP-2 expression was profoundly greater raised in hypertrophic scars and keloids than non-scarred tissue of patients [64]. Pyoderma gangrenosum, sweet’s syndrome, and hidradenitis suppurativa are chronic inflammatory skin diseases that surged not only cytokine expression (such as IL-1β, IL-17, IL-23, TNF-α), but also MMP-2, and MMP-9 over-production [65,66,67]. Furthermore, *P. acnes* influenced rising TNF-α, pro-MMP-2 mRNA and protein expression, which caused acne skin [68].

MMP production is related to elastin fibre fragmentation. The secretion of MMPs amplified the matrix remodelling process in hidradenitis suppurativa [66]. Especially, MMP-2 co-localized with keratinocytes, fibroblasts and macrophages cells in the dermis, sweat glands, hair follicles, and sinus tracts [69]. In addition, the reduction of MMP-2 expression could restore collagen types I and III that resulted in accelerating healing in wounds [22]. Interestingly, many medical plants can effectively activate the process of collagen biosynthesis [70,71]. Accordingly, the MMP-2 inhibitory capability of mushroom extracts may interrupt inflammatory skin problems and scar formation. However, the elevation of MMP-2 activity has a positive effect on degrading ECM in the initial state of the wound healing process which gave a benefit for the skin lesion [72].

Among all extracts, CW could give antioxidant potential and inhibit the production of inflammatory cytokines and MMP-2, which was comparable to the standard. Thus, it was responsible for pharmaceutical and cosmeceutical application as a wound healing product.

## 3. Materials and Methods

### 3.1. Materials

A sample of *D. indusiata* (Chinese species) in the form of a peach-shaped fruiting body was collected from Maerim’s bamboo mushroom farm, Chiang Mai, Thailand, in August 2020. Herbarium voucher specimens of *D. indusiata* sample (PNPRDU63032) was deposited in the Pharmaceutical and Natural Products Research and Development Unit (PNPRDU), Chiang Mai University, Chiang Mai, Thailand. The *D. indusiata* aqueous extracts were extracted and obtained from PNPRDU, Chiang Mai University, Chiang Mai, Thailand. The extracts of *D. indusiata* were divided into 3 types: peel and green mixture (PGW), core (CW), and whole mushroom (WW). The extraction method follows a patent pending process. The normal human fibroblasts immortalized by *hTRT* (OUMS-36T-2) was purchased from JCRB Cell Bank (Osaka, Japan) and RAW 264.7 (Mus musculus, mouse, macrophage) was provided by Assist. Prof. Korawinwich Boonpisuttinan (RMUTT, Thailand). Trypsin, Dulbecco’s Modified Eagle Medium (DMEM), Fetal bovine serum (FBS), penicillin, and streptomycin were obtained from Gibco (Grand Island, NY, USA). Trolox, 2,2-diphenyl-1-picrylhydrazyl (DPPH), 2,2-azino-bis (3–ethylbenzothiazoline–6-sulfonic acid) (ABTS) diammonium salt, ferrous sulfate, L-ascorbic acid, concanavalin A, ethylenediaminetetraacetic acid (EDTA), diclofenac sodium, and lipopolysaccharides (LPS) from *Escherichia coli* O55:B5 were purchased from Sigma Chemical Co. (St. Louis, MO, USA). All other chemical substances were of analytical grade.

### 3.2. Identification of Dictyophora indusiata Using Polymerase Chain Reaction (PCR) Based on Ribosomal DNA Internal Transcribed Spacers (rDNA-ITS)

#### 3.2.1. Defining Specific Primers

The sequences of rDNA in ITS regions of *D. indusiata* were designed using the basic local alignment search tool (BLAST) from the national center for biotechnology information (NCBI) (http://www.ncbi.nlm.nih.gov/blast/blast.cgi, accessed on 14 October 2021).

#### 3.2.2. DNA Extraction and PCR Reactions

The dry sample was grinded into the powder and performed DNA extraction using NucleoSpin^®^ Plant II (Macherey-Nagel, Germany), according to the instructions of the manufacturer. The PCR solution with a volume of 20 µL contained 4 µL of 5× HOT FIREPol^®^ Blend Master Mix (Solid BioDyne, Estonia), 2 µL of 10 µmol of primer, and 13 µL of distillated water. The PCR reactions were performed as start PCR with 10 min initial denaturation at 95 °C, 30 cycles of 30 s at 95 °C, 30 s at 59 °C, and 30 s at 68 °C, followed by a final extension of 5 min at 68 °C.

#### 3.2.3. Sequencing and Phylogenetic Analysis

The PCR products were determined by 2.2% agarose gel electrophoresis with 100 bp DNA ladder (Solid BioDyne, Estonia), and RedSafe™ (iNtRON, Korea). The gels were examined by a gel documentation system (Syngene, USA). PCR products were sequenced by Bio Basic Asia Pacific, Singapore. The sequences were aligned using Chromas, and performed Clustal Omega program for the multiple alignment [73]. Nucleotide sequences were compared using BLAST analysis in GenBank databases, followed by the construction of phylogenetic trees using Molecular Evolutionary Genetics Analysis (MEGA) X [74].

### 3.3. Determination of Bioactive Compounds

#### 3.3.1. Total Phenolic Content

The total phenolic contents in the extracts were determined by Folin-Ciocalteu colourimetric method, as previously described [42]. The mushroom extracts (12.5 µL), water (50 µL), and Folin-Ciocalteu reagent (12.5 µL) were added into 96-well plates and incubated for 6 min. After that, the mixture was neutralized by 10% w/v saturated sodium bicarbonate solution (125 µL) and distilled water (100 µL) which kept in the darkness at room temperature for 90 min. Gallic acid in different concentrations (20–800 µg/mL) was used as the standard phenolic substance. Total phenolic content was expressed as mg of gallic acid equivalents per g of dry extract (mg GAE/g extract).

#### 3.3.2. Total Flavonoid Content

Total flavonoid contents of the extracts were performed using aluminum chloride colourimetric assay, following the previous study [42]. The extracts (250 µL) were mixed with distilled water (1250 µL) in test-tube. Then, 5.0% NaNO_2_ solution (75 µL) was added and incubated at room temperature for 10 min. AlCl_3_·6H_2_O (150 µL) was subsequently added and incubated for another 10 min. The mixture was reacted with 1 M of NaOH (500 µL) and distilled water (275 µL) before the absorbance measurement. Different concentrations of catechin (0.01–0.32 mg/mL) were used in the standard calibration curve for total flavonoid determination. Total flavonoid content was expressed as mg of catechin equivalents per g of dry weight of each extract (mg CE/g extract).

#### 3.3.3. Total Polysaccharide Content

Total polysaccharide contents of the extracts were measured by Anthrone-Sulfuric acid method, as previous described [42]. The samples (250 µL) were reacted with 750 µL of Anthrone sulphate solution. The mixture was heated at 37 °C for 15 min and then cooled down for another 15 min. D-glucose monohydrate (in the range of 0.1–0.6 mg/mL) was used as a standard polysaccharide substance. The results of polysaccharide content were represented as µg of glucose equivalents per g of dry weight of each extract (µg GE/g extract).

#### 3.3.4. Quantitative Analysis of Phenolic and Flavonoid by Liquid Chromatography–Electrospray Ionization/Mass Spectrometry (LC-ESI/MS)

The extracts (10 mg) were dissolved in ethanol and filtrated through a 0.45 µm syringe filter into a vial bottle. The analysis was performed according to the already reported methods with some modifications [47,75] and analyzed using an Agilent 1260 Infinity II series, coupled with an electrospray ion (ESI) quadrupole mass spectrometry 6130 (Agilent Tech., Santa Clara, CA, USA). Reverse-phase column chromatography was performed using the Restek Ultra C18 column (250 × 4.6 mm, 4.6 mm, 5 µm) (Restek, Bellefonte, PA, USA). The column was maintained at 30 °C. The gradient elution was carried out using the 5% formic acid as a solvent A and acetonitrile: H_2_O: formic acid (85: 10: 5) as a solvent B with a linear gradian elution as 0–8 min, 80% A; 8–24 min, decreased A to 25%; 24–28 min, 25% A; 28–34 min, increased A to 70%; 34–36 min, increased A to 80%; 36–45 min, 80% A. The injection volume for all samples was 5 µL which was monitored at the flow rate of 0.5 mL ml/min. The MS was operated in the negative selected ion monitoring (SIM) as the following condition: dying gas (N2) flow, 12 L/min; dying gas temperature, 350 °C; nebulizer pressure, 60 psi; capillary voltage, 3000 V; fragmentor voltage, 70 V; and the full scan spectra from 100 to 1200 m/z with 250 ms/spectrum. The spectra were processed using Open Lab software (Agilent Tech., Santa Clara, CA, USA).

### 3.4. Antioxidant Assay of Dictyophora indusiata Aqueous Extracts

#### 3.4.1. DPPH Radical Scavenging Activity

The DPPH radical scavenging activity of PGW, CW, and WW was determined using DPPH by following the modified method [76]. The extracts were diluted with distilled water in the range of 0.0032–10 mg/mL. L-ascorbic acid was used as the positive control. The percentages of the DPPH scavenging activity were calculated by Equation (1), where Abs _Control_ is the absorbance of DPPH solution, and Abs _Sample_ is the absorbance of DPPH radicals reacted with extracts or standard: (1)$$\mathrm{DPPH scavenging activity}(\%)=\frac{\left({\mathrm{Abs}}_{\mathrm{Control}}-{\mathrm{Abs}}_{\mathrm{Sample}}\right)}{{\mathrm{Abs}}_{\mathrm{Control}}}\times 100$$

The linear relationship between the concentration of samples and the percentages of DPPH radical scavenging activity was plotted and calculated to the concentration providing 50% scavenging activity (SC_50_) (mg/mL).

#### 3.4.2. ABTS^+^ Scavenging Activity

The ABTS^+^ assay was determined by the previous method [76]. Briefly, ABTS^+^ solution was diluted with methanol to reach the absorbance of 0.68–0.72 at 734 nm. Then, 290 µL of ABTS^+^ solution reacted with 10 µL of different concentrations of the extracts (in the range of 0.625–30 mg/mL) for 10 min. The inhibitory concentration of samples was determined from a standard curve of Trolox. Then, the linear relationship between the concentration and the absorbance of samples was evaluated to obtain the SC_50_ (mg/mL).

#### 3.4.3. Metal Chelating Activity

The metal chelation by the extracts was determined using the inhibition of ferrozine–ferrous ion (Fe^2+^) complex formation, as previous described [42]. EDTA standard was used as the positive control. The percentages of chelating activity were calculated by Equation (2), Abs _Control_ is the absorbance of ferrozine–Fe^2+^ complex, and Abs _Sample_ is the absorbance of the complex reacted with extracts or standard:
(2)$$\mathrm{Chelating activity}(\%)=\frac{\left({\mathrm{Abs}}_{\mathrm{Control}}-{\mathrm{Abs}}_{\mathrm{Sample}}\right)}{{\mathrm{Abs}}_{\mathrm{Control}}}\times 100$$

The linear relationship between the concentration of samples and the percentages of chelating activity was plotted and calculated to the concentration providing 90% chelating activity (MC_90_) (mg/mL).

#### 3.4.4. Ferric Reducing Antioxidant Power (FRAP) Assay

The reducing power of extracts from *D. indusiata* was determined by FRAP assay, using the reported method [42]. FRAP reagent was prepared by 300 mmol/L acetate buffer, 20 mmol/L ferric chloride, and 10 mmol/L 2,4,6-tripyridyl-5-triazine in the ratio 10:1:1 (v:v:v). Then, 280 µL of FRAP reagent reacted with the extracts (in the range of 0.625–30 mg/mL) for 30 min at room temperature. Antioxidant potential of the extracts was calculated from the standard curve of ferrous sulfate. The result was represented as µM Fe^2+^/g extract.

### 3.5. Anti-inflammatory Activity of Dictyophora indusiata Aqueous Extracts

#### 3.5.1. Cell Culture

The RAW 264.7 macrophages were cultured in DMEM, supplemented with 10% (*v*/*v*) FBS, D-glucose (4500 mg/L), penicillin (100 U/mL) and streptomycin (100 mg/mL). The cells were incubated in a temperature-controlled and humidified incubator (CCL-050B-8, Esco^®^, Singapore) with 5% CO_2_ at 37 °C until the subcultures reached a confluence of 80%.

#### 3.5.2. Determination of Non-Cytotoxic Concentration by the Sulforhodamine B (SRB) Assay on Macrophages

The samples were tested for non-cytotoxic concentration on the RAW 264.7 macrophages by the SRB assay, the method was used with some modifications [77]. The cells (1 × 10^5^ cells/mL) were plated in 96-well plates and were incubated for 24 h. The cells were exposed to diclofenac sodium and the extracts (in the range of 0.0001–1 mg/mL) for another 24, 48, and 72 h, respectively. Then, the adherent cells were fixed in situ, washed, and dyed with SRB. The bound dye was solubilized, and the absorbance was measured at 515 nm. The percentages of cell viability were calculated by Equation (3), where Abs denotes absorbance:(3)$$\mathrm{Cell viability}(\%)=\frac{\left({\mathrm{Abs}}_{\mathrm{Sample}}-{\mathrm{Abs}}_{\mathrm{blank}}\right)}{\left({\mathrm{Abs}}_{\mathrm{Control}}-{\mathrm{Abs}}_{\mathrm{blank}}\right)}\times 100$$

#### 3.5.3. Determination of NO, IL-1, IL-6, and TNF-α Levels

NO, IL-1, IL-6 and TNF-α Levels were determined as previously described with slight modification [78]. The RAW 264.7 cells (1 × 10^5^ cells/mL) were cultured in 96-well plates for 24 h and treated with the samples (PGW, CW, and WW), and diclofenac sodium standard for 1 h. The treated cells were induced by 1 µg/mL of LPS for 24, 48, and 72 h, respectively. LPS was used as the positive control and diclofenac sodium was used as the negative control. After incubation, the supernatants were collected and determined for nitrite by using the Griess reagent kit (Thermo Fisher Scientific, Waltham, MA, USA), for IL-1β, IL-6, and TNF-α levels using ELISA kits (PEPROTECH, USA), according to the instructions of the manufacturer.

### 3.6. Collagen Stimulating Activity of Dictyophora indusiata Aqueous Extracts

#### 3.6.1. Cell Culture

The *hTRT* fibroblasts were cultured in DMEM, supplemented with 10% (*v*/*v*) FBS, penicillin (100 U/mL) and streptomycin (100 mg/mL). The cells were incubated in a temperature-controlled and humidified incubator (CCL-050B-8, Esco^®^, Singapore) with 5% CO_2_ at 37 °C until the subcultures reached a confluence of 80%.

#### 3.6.2. Determination of Non-Cytotoxic Concentration by the Sulforhodamine B (SRB) Assay on Fibroblasts

The samples were tested for non-cytotoxic concentration on the *hTRT* fibroblasts by the SRB assay, the method was used with some modifications [77]. The cells (1 × 10^5^ cells/mL) were plated in 96-well plates and were incubated for 24 h. The fibroblasts were treated with different concentrations of L-ascorbic acid, concanavalin A, and the extracts (in the range of 0.0001–1 mg/mL) for another 24 h. After that, the adherent cells were fixed in situ, washed, and dyed with SRB. The bound dye was solubilized, and the absorbance was measured at 515 nm.

#### 3.6.3. Gelatinolytic Activity (Zymography) of MMP-2 Inhibition

MMP-2 level was determined according to the previous method [76]. The fibroblasts (5 × 10^5^ cells/mL) were cultured in 6-well plates for 24 h and maintained in the culture medium without FBS for another 24 h. Then the cells were treated with the samples (PGW, CW, and WW), concanavalin A (negative control) and L-ascorbic acid (positive control) standard, respectively, and incubated for 48 h. The cultured supernatants were collected and tested on 10% SDS-PAGE gels containing 1 mg/mL of gelatin. The gels were stained with 0.5% Coomassie Brilliant Blue G-250, scanned by a gel documentation system (Gel Doc^TM^ EZ Gel, Bio-Rad Laboratories, Hercules, CA, USA) and analysed using the Image Lab^TM^ software 5.1 (Bio-Rad Laboratories, USA). The volume (intensity) multiplied by the number of pixels of the bands on the gel was determined as the relative MMP-2 content (intensity unit). The percentages of MMP-2 inhibition relative to the control (the untreated systems) were calculated by Equation (4):(4)$$\mathrm{MMP-2 inhibition}(\%)=100-[\frac{\left({\mathrm{MMP-2 content}}_{\mathrm{Sample}}\right)}{{\mathrm{MMP-2 content}}_{\mathrm{Control}})}\times 100]$$

### 3.7. Statistical Analysis

All experiments were operated in at least triplicate for each test. Data was expressed as means ± standard deviation (SD). One-way ANOVA, followed by LSD’s post hoc test was used to analyse the significant differences using SPSS 23.0 software (SPSS Inc., Chicago, IL, USA). A value of *p* < 0.05 was considered statistically significant.

## 4. Conclusions

Since there is no publication of the in vitro wound healing effect of immature (egg) bamboo mushroom extracts, the anti-inflammatory and collagen stimulating (MMP-2 inhibition) activities for the wound healing were carried out in this study. The aqueous extracts from the immature stage of core *D. indusiata* mushroom showed the considerable wound healing effect not only with the highest antioxidant activity, the comparable anti-inflammation via the reduction of cytokines (NO, IL-1, IL-6, and TNF-α) secretion from LPS-induced macrophage cells to the standard diclofenac, but also the promising MMP-2 inhibition through the determination of gelatinolytic activity on fibroblasts cells which were responsible for inflammation, cell proliferation, and tissue remodelling stages in the wound healing process, respectively. Over-production of inflammatory cytokines and MMP-2 lead to wound healing impairment, chronic skin inflammatory diseases, and scar formation. Therefore, CW which provided the high content of catechin, other polyphenolic compounds, and the effective anti-inflammation and MMP-2 inhibition activity could have the promising in vivo wound healing effect could be a candidate to develop as pharmaceutical and/or cosmeceutical wound healing active ingredients. Hence, the in vivo study of CW extract should be further investigated in animal and volunteers with wound injury for the honorable result. The animal and human ethical approval are currently on the application process.

## Figures and Tables

**Figure 1 jof-07-01100-f001:**
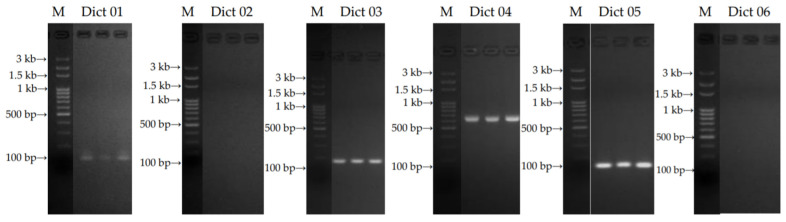
PCR products of the sample from the 6 specific primers (Dict 01, 02, 03, 04, 05 and 06 primers); marker lane (M).

**Figure 2 jof-07-01100-f002:**
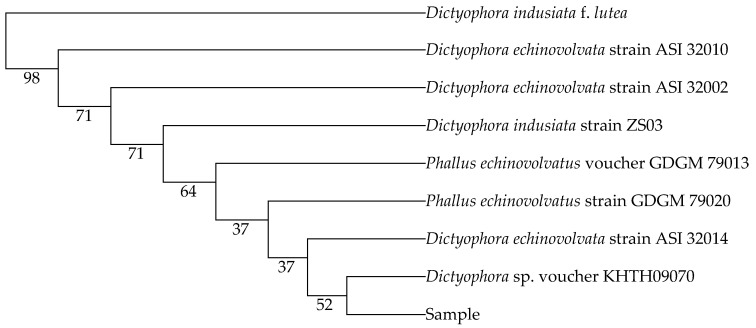
The phylogenetic tree of the sample resulting from Dict 03 primer. Neighbor joining: bootstrap analysis (1000 replications) by MEGA X.

**Figure 3 jof-07-01100-f003:**
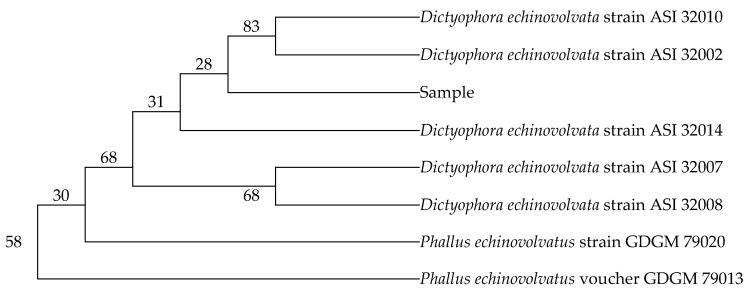
The phylogenetic tree of the sample resulting from Dict 04 primer. Neighbor joining: bootstrap analysis (1000 replications) by MEGA X.

**Figure 4 jof-07-01100-f004:**
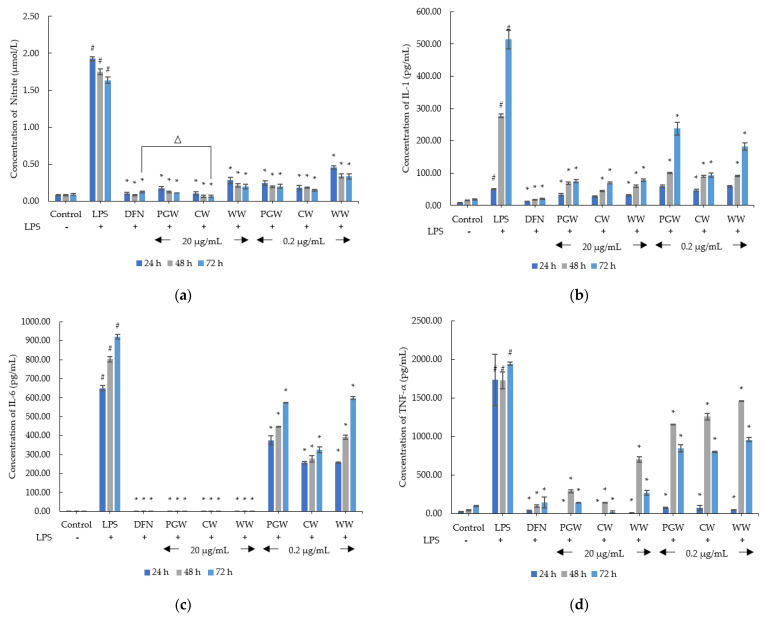
(**a**) The nitrite concentration; (**b**) interleukin-1β (IL-1β) concentration; (**c**) interleukin-6 (IL-6) concentration (**d**) tumour necrosis factor-α (TNF-α) concentration treated by *Dictyophora indusiata* aqueous extracts, peel and green mixture (PGW), core (CW), and whole (WW) at each concentration (20 and 0.2 µg/mL) for 24 h, 48 h, and 72 h. DMEM (control), lipopolysaccharides (LPS) (positive control) and diclofenac sodium (DFN) (negative control), respectively. Significant differences are indicated as ^#^
*p* < 0.05 (compared to the control), * *p* < 0.05 (compared to the positive control), and ^∆^
*p* < 0.05 (compared to the negative control).

**Figure 5 jof-07-01100-f005:**
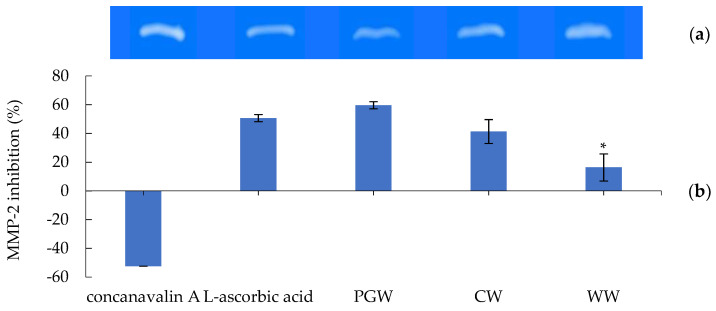
(**a**) Zymograms; (**b**) the percentages of matrix metalloproteinase 2 (MMP-2) inhibition. The comparison of the gelatinolytic activity of MMP-2 inhibition on human skin fibroblasts between aqueous extracts of *Dictyophora indusiata* at 1 mg/mL, L-ascorbic acid (positive control), and concanavalin A (negative control), peel and green mixture (PGW), core (CW), whole (WW). Significant differences are indicated as * *p* < 0.05 (compared to the positive control).

**Table 1 jof-07-01100-t001:** Primer sequences for sample amplification.

Primer Code Name	Primer Sequences (5′-3′)	Size of Product (bp)	References
Dict 01: forward	AGGCCTCTCGAAAGAGGGTC	20	[23]
Dict 01: reverse	TCATCGATGCGAAAGCCAAG	20	
Dict 02: forward	TCGCGCGTGTCAGTGAAATA	20	[23]
Dict 02: reverse	CCAAGTCCGAAAGGGGTCTC	20	
Dict 03: forward	TGCCTGTTTGAGTGTCGTGA	20	[23]
Dict 03: reverse	ACGGACGACGCAAGACTTAT	20	
Dict 04: forward	GGAAGTAAAAGTCGTAACAAGG	22	[23]
Dict 04: reverse	TCCTCCGCTTATTGATATGC	20	
Dict 05: forward	AGGAGCATGCCTGTTTGAGT	20	[24]
Dict 05: reverse	TGGAAACCTCGCCGATGAAT	20	
Dict 06: forward	GTCATGAACGCCCGTTTCTC	20	[24]
Dict 06: reverse	ACCCTCCTTCCGATGAGACT	20	

**Table 2 jof-07-01100-t002:** Nucleotide sequences of the sample.

Specific Primer	Sequences (5′-3′)	Size (bp)
Dict 03	TTACCGAAGGAGGCAGGACTAACAAGTTCGGAAGGGGGGTAAAGGGGAAGGG GGACCTTCGCCGATGAATTTGAAGACGAGCCTTCGACCGCAGGGGGATTCGAG GGCAAGACCGTCCAAGTCCGAAAAAAGGAGAAATCCGTTAA	146
Dict 04	TTCCTTCCTTTCCTCCGCTTATTGATATGATTAAGTTGGGCGGGTAATCCTGCCTG ATTTGAGGTCAAGGCGTATAATGAATGACGGAACGAGAAGCCCACCCCGCCCT TTTTTTTCCCCCCAGGACGAAGCAAGACTTATCAAGTTTGGATGGGGGGTAAAG GGGAAGGGGGACCTTCGCCGATGAATTTGAAGACGAGCCTTCGACCGCAGGGG GATTGGAGGGCAAGACCGTCCAAGTCAAGAAAAAAGGGAGAAATCCTTTTTTT CGATGAGATTTCACGACACTCAAACAGGCATGCTCCTCGGAATACCGAGGAGC GCAAGATGCGTTCAAAGATTCGATGATTCATTGAATTCTGCAATTCACATTACG TTTCGCGCGTTCGCGGCGTTCTTCATCGATGCGAAAGCCAAGAGATCCGTTGTT GAAAGTTGTGTTTCGATTTTTATTTCACTGACACGCGCGAGACTGCGAGGCGTTT GTGAAAGACGGGAGGGGCCAAGCCTCTTTCGAGAGGCCTCTCCCAGAGTGCAC GGAGGTGTCGGTCGGGGAGAGAGAGCGCGTCTCCCCCCCGGATGATAAATCGG CAATGATCTCCGCAGTACAGAG	613

**Table 3 jof-07-01100-t003:** Nucleotide identity between the nucleotide sequences of the sample resulting from Dict 03 primer and the fungi species.

Species	GenBank Accession Number	Nucleotide Identity (%)
*Dictyophora indusiata* strain ZS03	MH464257.1	98
*Dictyophora echinovolvata* strain ASI 32010	AF324167.2	98
*Dictyophora echinovolvata* strain ASI 32002	AF324164.2	98
*Phallus echinovolvatus* voucher GDGM 79013	MN613536.1	97
*Phallus echinovolvatus* strain GDGM 79020	MN523216.1	97
*Dictyophora echinovolvata* strain ASI 32014	AF324168.2	97
*Dictyophora* sp. voucher KHTH09070	MG678511.1	97
*Dictyophora indusiata* f. *lutea*	HQ414538.1	95

**Table 4 jof-07-01100-t004:** Nucleotide identity between the nucleotide sequences of the sample resulting from Dict 04 primer and the fungi species.

Species	GenBank Accession Number	Nucleotide Identity (%)
*Phallus echinovolvatus* voucher GDGM 79013	MN613536.1	97
*Phallus echinovolvatus* strain GDGM 79020	MN523216.1	97
*Dictyophora echinovolvata* strain ASI 32014	AF324168.2	97
*Dictyophora echinovolvata* strain ASI 32007	AF324165.2	97
*Dictyophora echinovolvata* strain ASI 32008	AF324166.2	97
*Dictyophora echinovolvata* strain ASI 32010	AF324167.2	95
*Dictyophora echinovolvata* strain ASI 32002	AF324164.2	95

**Table 5 jof-07-01100-t005:** Bioactive contents of 3 types of *Dictyophora indusiata* aqueous extracts.

Sample	Total Phenolic Contents (mg GAE/g Extract)	Total Flavonoid Contents (mg CE/g Extract)	Total Polysaccharide Contents (µg GE/g Extract)
PGW	2.55 ± 0.36	0.05 ± 0.02	2.22 ± 0.29
CW	2.05 ± 0.08	0.01 ± 0.02	1.20 ± 0.14
WW	1.89 ± 0.17	0.02 ± 0.01	1.65 ± 0.09

Each value is expressed as mean ± SD (*n* = 3); peel and green mixture (PGW); core (CW); whole (WW); mg GAE/g extract = mg of gallic acid equivalents per g of dry extract; mg CE/g extract = mg of catechin equivalents per g of dry weight of each extract; µg GE/g extract = µg of glucose equivalents per g of dry weight of each extract.

**Table 6 jof-07-01100-t006:** Polyphenol compounds of 3 types of *Dictyophora indusiata* aqueous extracts.

Polyphenol Compounds (mg/g Extract)	PGW	CW	WW
Catechin	3.481 ± 0.001	68.761 ± 0.010	2.934 ± 0.010
*p*-Coumaric acid	3.887 ± 0.043	7.931 ± 0.939	4.066 ± 0.079
Rutin	0.476 ± 0.092	2.502 ± 0.008	3.290 ± 0.027
Rosmarinic acid	3.270 ± 0.014	0.235 ± 0.009	0.178 ± 0.006
Quercetin	0.754 ± 0.007	0.055 ± 0.004	0.833 ± 0.013
Naringenin	Nd	0.516 ± 0.003	0.209 ± 0.000
Epigallocatechin gallate (EGCG)	Nd	0.767 ± 0.004	Nd

Each value is expressed as mean ± SD (*n* = 3); peel and green mixture (PGW); core (CW); whole (WW); Nd = Not determined.

**Table 7 jof-07-01100-t007:** Antioxidant activities of 3 types of *Dictyophora indusiata* aqueous extracts.

Sample	DPPH Scavenging Activity (SC_50_, mg/mL)	ABTS Scavenging Activity (SC_50_, mg/mL)	Metal Chelating Activity (MC_90_, mg/mL)	FRAP Reducing Power (µM Fe^2+^/g Extract)
PGW	2.51 ± 0.05 *	8.52 ± 0.09	32.18 ± 0.23 *	6.18 ± 0.08
CW	1.54 ± 0.02 *	6.51 ± 0.08	5.11 ± 0.45 *	2.13 ± 0.09
WW	3.59 ± 0.02 *	11.81 ± 0.09	34.58 ± 0.39 *	3.00 ± 0.03
EDTA	Nd	Nd	0.10 ± 0.09	Nd
L-ascorbic acid	0.26 ± 0.02	Nd	Nd	Nd

Each value is expressed as mean ± SD (*n* = 3); peel and green mixture (PGW); core (CW); whole (WW). Ethylenediaminetetraacetic acid (EDTA); SC_50_ = the concentration providing 50% scavenging activity (mg/mL); MC_90_ = the concentration providing 90% chelating activity (mg/mL); µM Fe^2+^/g extract = µM of ferrous ion per g of extract; Nd = Not determined. Significant differences are indicated as * *p* < 0.05 (compare to the standard substances).

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
