# Peer review of "High Efficiency In Vitro Wound Healing of Dictyophora indusiata Extracts via Anti-Inflammatory and Collagen Stimulating (MMP-2 Inhibition) Mechanisms"

_jof, 2021, doi:10.3390/jof7121100_

Round 1

Reviewer 1 Report

Dear Authors,

The MS in current form can be accepted for publishing.

Author Response

REVIEWER 1

  1. The MS in current form can be accepted for publishing.

Response: Thank you for positive comment.

Reviewer 2 Report

Dear Authors

Good day!

This manuscript has a lot of information to share with its readers. Despite that fact It seems that you have not even done a last checking about your manuscript and send it without removing highlights.

This manuscript's English needs to be extensively corrected, there was so many missed spaces, wrong capitalization, wrong degree sign (Line 469), so many nonessential abbreviation which they have not even repeated once in the manuscript. 

-Please remove the brakes from title in line 4.

-Abstract is too weak and it should change in whole

-line 82 then suddenly changed to line 103???

-line 110. you have to mention some of those various bio activities!

-Figure 2 and Figure 3. Please add proper lead information to figures for clarity of interpretation of the phylogenetic tree's numbers

-Table 6. P-coumaric acid the P should not be capital!!!

-Figure 5. Concanavalin A should not start with capital C.

More importantly, introduction has no storyline and does not specify the reason why you submitted your manuscript to JoF and why not to a medical journal! so clear your thought and be more specified with the journal you are submitting your work too. 

Please cut the medical information and pathways, just focus on special bioactivity of the fungi and the specific effect of it. 

all this suggestions are for the sake of your readers and easy understanding of your work.

regards

Author Response

REVIEWER 2

  1. This manuscript has a lot of information to share with its readers. Despite that fact It seems that you have not even done a last checking about your manuscript and send it without removing highlights.

Response: Thank you for your comment. Since this submission is the re-submit version to this journal so we have to declare what we have changed or updated in this manuscript. However, all highlights have been removed from the revised manuscript (in the latest version) using the track change instead.

  1. This manuscript's English needs to be extensively corrected, there was so many missed spaces, wrong capitalization, wrong degree sign (Line 469), so many nonessential abbreviation which they have not even repeated once in the manuscript.

Response: Thank you for your comment. We are sorry for the mistake. The revised manuscript has been thoroughly checked and corrected and the degree sign (Line 449) has been modified. The completed terms of FGF, VEGF, and PDGF have been already mentioned in the introduction part. Therefore, these abbreviations have been addressed as “…growth factors (FGF2, monocyte chemoattractant pro-tein-1 (MCP-1), macrophage inflammatory protein-1α (MIP-1α), VEGF-A, and PDGF-BB) levels”, please see on page 7 (lines 227-229).

  1. Please remove the brakes from title in line 4.

Response: Thank you for your comment. Unfortunately, we cannot find the brake in line 4. Please let us know if we misunderstand.

  1. Abstract is too weak and it should change in whole

Response: Thank you for your kindness. The abstract has been modified and rearranged. The abstract: “Dictyophora indusiata is widely used as not only traditional medicine, functional foods, but also, skin care agents. The egg stage of bamboo mushroom was selected to investigate biological activities. Nucleotide sequencing was evaluated for molecular identification of this mushroom. The mushroom was divided into 3 parts: peel and green mixture (PGW), core (CW), and whole mushroom (WW) for preparing water extraction, quantitative analysis of bioactive compounds, and determination of antioxidant activity. Wound healing effect was introduced by the anti-inflammatory determination via the levels of cytokine releasing from macrophages, and the collagen stimulation activity on fibroblasts by matrix metalloproteinase-2 (MMP-2) inhibitory activity. All D. indusiata extracts showed good antioxidant potential, significantly anti-inflammatory activity in the decreasing of the nitric oxide (NO), interleukin-1 (IL-1), interleukin-1 (IL-6), and tumour necrosis factor-α (TNF-α) secretion from macrophage cells (p < 0.05), and the effective collagen stimulation via MMP-2 inhibition. In particular, CW extract containing high content of catechin which could significantly suppress NO secretion better than the standard anti-inflammatory drug diclofenac and their MMP-2 inhibition was comparable to L-ascorbic acid (p > 0.05). These findings support that CW of D. indusiata could be an essential natural active ingredient for health promoting application as skin wound healing pharmaceutical products.” has been reconstructed to Dictyophora indusiata or Phallus indusiatus is widely used as not only traditional medicine, functional foods, but also, skin care agents. The egg stage of bamboo mushroom or bamboo fungus was divided into 3 parts: peel and green mixture (PGW), core (CW), and whole mushroom (WW). The aqueous extracts were investigated for their nucleotide sequencing, biological compound   contents, and biological activities. Wound healing effect was introduced by the anti-inflammatory determination via the levels of cytokine releasing from macrophages, and the collagen stimulation activity on fibroblasts by matrix metalloproteinase-2 (MMP-2) inhibitory activity. All D. indusiata extracts showed good antioxidant potential, significantly anti-inflammatory activity in the decreasing of the nitric oxide (NO), interleukin-1 (IL-1), interleukin-1 (IL-6), and tumour necrosis factor-α (TNF-α) secretion from macrophage cells (p < 0.05), and the effective collagen stimulation via MMP-2 inhibition. In particular, CW extract containing high content of catechin which could significantly suppress NO secretion better than the standard anti-inflammatory drug diclofenac and their MMP-2 inhibition was comparable to L-ascorbic acid (p > 0.05). These findings support that CW of D. indusiata could be an essential natural active ingredient for health promoting application as skin wound healing pharmaceutical products.”, please see on page 1 (lines 23-37).

  1. Line 82 then suddenly changed to line 103???

Response: Thank you for your comment. We are sorry for the mistake. The line numbers have been thoroughly checked and corrected.

  1. Line 110. you have to mention some of those various bio activities!

Response: Thank you for your comment. Theses bioactivities have been already described on page 2 (lines 50-54).

  1. Figure 2 and Figure 3. Please add proper lead information to figures for clarity of interpretation of the phylogenetic tree's numbers

Response: Thank you for your comment. The phylogenetic trees have been obtained from neighbor joining (NJ) method with 1000 bootstrap replications using Molecular Evolutionary Genetics Analysis (MEGA) X. However, the phylogenetic tree's numbers have been modified as follows:

Dictyophora indusiata f. lutea

Dictyophora echinovolvata strain ASI 32010

98

Dictyophora echinovolvata strain ASI 32002

71

Dictyophora indusiata strain ZS03

                        71

Phallus echinovolvatus voucher GDGM 79013

                                      64

Phallus echinovolvatus strain GDGM 79020

                                                  37

Dictyophora echinovolvata strain ASI 32014

                                                               37

Dictyophora sp. voucher KHTH09070

                                                                       52

Sample

Figure 2. The phylogenetic tree of the sample resulting from Dict 03 primer. Neighbor joining: bootstrap analysis (1000 reps) by MEGA X

Dictyophora echinovolvata strain ASI 32010

83

Dictyophora echinovolvata strain ASI 32002

28

Sample

31

Dictyophora echinovolvata strain ASI 32014

68

Dictyophora echinovolvata strain ASI 32007

30

68

Dictyophora echinovolvata strain ASI 32008

58

Phallus echinovolvatus strain GDGM 79020

Phallus echinovolvatus voucher GDGM 79013

Figure 3. The phylogenetic tree of the sample resulting from Dict 04 primer Neighbor joining: bootstrap analysis (1000 reps) by MEGA X

  1. Table 6. P-coumaric acid the P should not be capital!!!

Response: Thank you for your comment. The word “P-coumaric acid” has been edited to “p-coumaric acid”, please see on Table 6.

  1. Figure 5. Concanavalin A should not start with capital C.

Response: Thank you for your comment. The word “Concanavalin A” has been changed to “concanavalin A”, please see on Figure 5.

  1. More importantly, introduction has no storyline and does not specify the reason why you submitted your manuscript to JoF and why not to a medical journal! so clear your thought and be more specified with the journal you are submitting your work too. Please cut the medical information and pathways, just focus on special bioactivity of the fungi and the specific effect of it. All these suggestions are for the sake of your readers and easy understanding of your work.

Response: Thank you for your kindness. The introduction had been modified as your suggestion. More information about the mushroom bamboo which related to JoF had been added and a storyline was introduced in the introduction part. The medical information had been deleted. The introduction “Skin is the largest organ of our body. The skin functions are to protect our organs from the external environment like microorganisms and allergens and maintain internal factors such as hormones, water balance, and temperature. [1]. In addition, skin damage can cause inflammation and infection called wound infection and septicaemia, respectively [2]. Usually, acute wounds are able to heal and resolve themselves promptly within a month [3]. In contrast, chronic wounds, for example, decubitus ulcers, venous ulcers, and diabetic wounds fail to heal appropriately [4,5]. Reducing tissue injury, epithelialization of living tissue, providing sufficient oxygen and nutrition, and preventing wounds’ dehydration, are the principle of the wound healing process [5-7]. The wound repairing process included 4 main approaches. Firstly, coagulation or hemostasis stage, fibrinogens from traumatic areas are converted from thrombin to fibrin. Fibrin clot formation is the crucial step in coagulation cascades to stop blood loss [8]. During this phase, the number of soluble mediators such as platelet-derived growth factor (PDGF), insulin-like growth factor-1 (IGF-1), epidermal growth factor (EGF), fibroblast growth factor (FGF), transforming growth factor-β (TGF-β), and vascular endothelial cell growth factor (VEGF) are released from platelets [9-11].  Secondly, inflammation phase, polymorphonuclear leukocytes (PMN) and macrophages migrate to the damaging areas to defend microbes, attract other macrophage cells, support cell proliferation, and secrete cytokines and protease (elastase and collagenase) [9,12]. Injured extracellular matrix (ECM) components are degraded by these two proteases. Cytokines containing interleukin-1 (IL-1) and tumour necrosis factor-α (TNF-α) activated proteases production and apoptosis in fibroblasts. Thirdly, the promotion of cell proliferation and restoration of the matrix, fibroblasts, endothelial cells, and keratinocytes are conducted, and inflammatory cells are eliminated. Growth factors and cytokines involving TGF-β, TGF-α, IL-1, IGF-1, FGF, PDGF, VEGF, keratinocyte growth factor (KGF), and connective tissue growth factor (CTGF) are synthesized to promote cell proliferation, develop new capillary formation, and produce new ECM components [13,14]. Subsequently, matrix metalloproteinases (MMPs) will remove injured matrix proteins and then fibroblasts release lysyl oxidase to link collagen at ECM in the scar forming. Finally, in tissue remodelling or scar maturation state, the initial scars are red and raised. Then, ECM components production in the scar and their degradation by proteases would reach the new equilibrium, and scar tissues will improve [8,9]. However, the most common wound-healing problem is the hypertrophic scar or keloid resulting from prolonged inflammation [7] and the action of MMP-2, MMP-3, and MMP-13 [8]. Moreover, inflammatory skin implications like atopic dermatitis particularly increased of IL-1α, IL-1β, TNF-α, and granulocyte-macrophage-colour-stimulating factor (GM-CSF) [15]. In acne lesions, IL-1 remarkably elevated around follicles in both the interfollicular and perifollicular epidermis of acne patients [16]. Thus, the suppression of the inflammatory cytokines improves skin structure and barrier functions in patients with skin diseases [17]. Moreover, MMP-2 inhibitory potential could restore the balance of collagen production [18]. Mushroom is rich in bioactive compounds, for example, polysaccharides, polyphenolics, vitamins, and has been used in the cosmeceutical industry for anti-ageing, moisturizing as well as skin lightening [19,20]. Many studies revealed that the fruiting body of Dictyophora indusiata showed various impressive activities such as anti-obesity [21], and neuroprotective effect for treating Alzheimer’s disease [22]. Furthermore, D. indusiata also has considerable bioactivities with cosmeceutical potentials such as antioxidant, anti-tyrosinase, antimicrobial properties [23-25]. However, most research papers reported various bioactivities of the fruiting body from D. indusiata, while the studies on the application of immature bamboo mushroom extracts as cosmeceutical or pharmaceutical applications were only a few. Therefore, this study aimed to investigate total phenolic, flavonoid, and polysaccharide contents, then further determine the antioxidant capabilities of aqueous extracts from 3 parts of D. indusiata mushroom. In addition, these extracts were assessed for wound healing activity using RAW 264.7 macrophages and hTRT fibroblasts which compared to the standard substances. The results from this study may consider as bioactive sources for pharmaceutical and/or cosmeceutical applications.” have been revised to “Mushroom is rich in bioactive compounds, for example, polysaccharides, polyphenols, vitamins, and has been used in the cosmeceutical industry for anti-ageing, moisturizing as well as skin lightening [1,2]. Dictyophora indusiata or Phallus indusiatus or bamboo mushroom is a fungus belonging to the family Phallaceae. The bioactive compositions in immature and mature stages of this mushroom are different. The egg or immature mushroom produce viscous mucilage and high phenolic contents. There has been used as a tablet binder and probiotic co-encapsulation [3,4]. On the other hand, the mature stage of this mushroom contains many bioactive compounds, for example, polysaccharides, amino acids, terpenoids, and alkaloids [5,6]. Many studies revealed that the fruiting body of Dictyophora indusiata showed various impressive activities such as anti-obesity [7], and neuroprotective effect for treating Alzheimer’s disease [8]. Furthermore, D. indusiata also has considerable bioactivities with cosmeceutical potentials such as antioxidant, anti-tyrosinase, antimicrobial properties [9-11]. Skin damage can cause inflammation and infection called wound infection and septicaemia, respectively [12]. Usually, acute wounds are able to heal and resolve themselves promptly within a month [13]. In contrast, chronic wounds, for example, decubitus ulcers, venous ulcers, and diabetic wounds fail to heal appropriately [14,15]. The wound repairing process included 4 main approaches. Firstly, Fibrin formation is the crucial step in coagulation cascades to stop blood loss [16]. During this phase, the number of soluble mediators such as platelet-derived growth factor (PDGF), insulin-like growth factor-1 (IGF-1), epidermal growth factor (EGF), fibroblast growth factor (FGF), transforming growth factor-β (TGF-β), and vascular endothelial cell growth factor (VEGF) are released from platelets [17-19]. Secondly, inflammation phase, macrophages migrate to the damaging areas to defend microbes, attract other macrophage cells, support cell proliferation, and secrete cytokines and protease (elastase and collagenase) [17,20]. Injured extracellular matrix (ECM) components are degraded by these two proteases. Cytokines containing interleukin-1 (IL-1) and tumour necrosis factor-α (TNF-α) activated proteases production and apoptosis in fibroblasts. Thirdly, the promotion of cell proliferation and restoration of the matrix, fibroblasts, endothelial cells, and keratinocytes are conducted. Growth factors and cytokines involving TGF-β, TGF-α, IL-1, IGF-1, FGF, PDGF, VEGF, keratinocyte growth factor (KGF), and connective tissue growth factor (CTGF) are synthesized to promote cell proliferation, develop new capillary formation, and produce new ECM components [21,22]. Subsequently, matrix metalloproteinases (MMPs) will remove injured matrix proteins and then fibroblasts release lysyl oxidase to link collagen at ECM in the scar forming. Finally, ECM components production and their degradation would reach the equilibrium, and scar tissues will improve [16,17]. However, the most common wound-healing problem is the  hypertrophic scar or keloid resulting from prolonged inflammation [23] and the action of MMP-2, MMP-3, and MMP-13 [16]. Moreover, inflammatory skin implications like atopic dermatitis particularly increased of IL-1α, IL-1β, TNF-α, and granulocyte-macrophage- colour-stimulating factor (GM-CSF) [24]. In acne lesions, IL-1 remarkably elevated around follicles in both the interfollicular and perifollicular epidermis of acne patients [25]. Thus, the suppression of the inflammatory cytokines improves skin structure and barrier functions in patients with skin diseases [26]. Moreover, MMP-2 inhibitory potential could restore the balance of collagen production [27]. [7][8][9-11]

However, most research papers reported various bioactivities of the fruiting body from D. indusiata, while the studies on the application of immature bamboo mushroom extracts as cosmeceutical or pharmaceutical applications were only a few. Therefore, this study aimed to investigate total phenolic, flavonoid, and polysaccharide contents, then further determine the antioxidant capabilities of aqueous extracts from 3 parts of D. indusiata mushroom. In addition, these extracts were assessed for wound healing activity using RAW 264.7 macrophages and hTRT fibroblasts which compared to the standard substances. The results from this study may consider as bioactive sources for pharmaceutical and/or cosmeceutical applications.”, please see on pages 1-2 (lines 42-94).

References:

  1. Taofiq, O.; González-Paramás, A.M.; Martins, A.; Barreiro, M.F.; Ferreira, I.C. Mushrooms extracts and compounds in cosmetics, cosmeceuticals and nutricosmetics—A review. Ind Crops Prod 2016, 90, 38-48.
  2. Wu, Y.; Choi, M.-H.; Li, J.; Yang, H.; Shin, H.-J. Mushroom cosmetics: the present and future. Cosmetics 2016, 3, 22.
  3. Srisuk, N.; Jirasatid, N. Characteristics co-encapsulation of Lactobacillus acidophilus with Dictyophora indusiata. Curr Res Nutr Food Sci 2020, 8, 1013.
  4. Burapapadh, K.; Changsan, N.; Sinsuebpol, C.; Saokham, P. An evaluation of Dictyophora indusiata mucilage as a binder in tablet formulations. Key Eng Mater 2021, 901, 22-27.
  5. Habtemariam, S. The chemistry, pharmacology and therapeutic potential of the edible mushroom Dictyophora indusiata (Vent ex. Pers.) Fischer (Synn. Phallus indusiatus). Biomedicines 2019, 7, 98.
  6. Wang, J.; Wen, X.; Zhang, Y.; Zou, P.; Cheng, L.; Gan, R.; Li, X.; Liu, D.; Geng, F. Quantitative proteomic and metabolomic analysis of Dictyophora indusiata fruiting bodies during post-harvest morphological development. Food Chem 2021, 339, 127884.
  7. Wang, W.; Song, X.; Zhang, J.; Li, H.; Liu, M.; Gao, Z.; Wang, X.; Jia, L. Antioxidation, hepatic-and renal-protection of water-extractable polysaccharides by Dictyophora indusiata on obese mice. Int J Biol Macromol 2019, 134, 290-301.
  8. Talebi, M.; Kakouri, E.; Talebi, M.; Tarantilis, P.A.; Farkhondeh, T.; İlgün, S.; Pourbagher-Shahri, A.M.; Samarghandian, S. Nutraceuticals-based therapeutic approach: recent advances to combat pathogenesis of Alzheimer’s disease. Expert Rev Neurother 2021, 21, 625-642.
  9. Oyetayo, V.; Dong, C.-H.; Yao, Y.-J. Antioxidant and antimicrobial properties of aqueous extract from Dictyophora indusiata. Open Mycol J 2009, 3.
  10. Sharma, V.K.; Choi, J.; Sharma, N.; Choi, M.; Seo, S.Y. In vitro anti‐tyrosinase activity of 5‐(hydroxymethyl)‐2‐furfural isolated from Dictyophora indusiata. Phytother Res: An International Journal Devoted to Pharmacological and Toxicological Evaluation of Natural Product Derivatives 2004, 18, 841-844.
  11. Hua, Y.; Yang, B.; Tang, J.; Ma, Z.; Gao, Q.; Zhao, M. Structural analysis of water-soluble polysaccharides in the fruiting body of Dictyophora indusiata and their in vivo antioxidant activities. Carbohydr Polym 2012, 87, 343-347.
  12. Percival, N.J. Classification of wounds and their management. Surgery (Oxford) 2002, 20, 114-117.
  13. Velnar, T.; Bailey, T.; Smrkolj, V. The wound healing process: an overview of the cellular and molecular mechanisms. J Int Med Res 2009, 37, 1528-1542.
  14. Grey, J.E.; Enoch, S. ABC of wound healing: pressure ulcers. Bmj 2006, 332.
  15. Pierce, P.D., MD, Glenn F; Mustoe, M., Thomas A. Pharmacologic enhancement of wound healing. Annu Rev Med 1995, 46, 467-481.
  16. Shih, B.; Garside, E.; McGrouther, D.A.; Bayat, A. Molecular dissection of abnormal wound healing processes resulting in keloid disease. Wound Repair Regen 2010, 18, 139-153.
  17. Schultz, G.S.; Sibbald, R.G.; Falanga, V.; Ayello, E.A.; Dowsett, C.; Harding, K.; Romanelli, M.; Stacey, M.C.; Teot, L.; Vanscheidt, W. Wound bed preparation: a systematic approach to wound management. Wound Repair Regen 2003, 11, S1-S28.
  18. Arisato, T.; Hashiguchi, T.; Sarker, K.; Arimura, K.; Asano, M.; Matsuo, K.; Osame, M.; Maruyama, I. Highly accumulated platelet vascular endothelial growth factor in coagulant thrombotic region. J Thromb Haemost 2003, 1, 2589-2593.
  19. Verheul, H.M.; Hoekman, K.; Lupu, F.; Broxterman, H.J.; Van Der Valk, P.; Kakkar, A.K.; Pinedo, H.M. Platelet and coagulation activation with vascular endothelial growth factor generation in soft tissue sarcomas. Clin Cancer Res 2000, 6, 166-171.
  20. Enoch, S.; Leaper, D.J. Basic science of wound healing. Surgery (Oxford) 2005, 23, 37-42.
  21. Buranasukhon, W.; Athikomkulchai, S.; Tadtong, S.; Chittasupho, C. Wound healing activity of Pluchea indica leaf extract in oral mucosal cell line and oral spray formulation containing nanoparticles of the extract. Pharm Biol 2017, 55, 1767-1774.
  22. Ruksiriwanich, W.; Khantham, C.; Linsaenkart, P.; Jantrawut, P.; Rajchasom, S. Optimization of placenta extraction for wound healing activity. Chiang Mai J Sci 2019, 46, 946-959.
  23. Menon, S.N.; Flegg, J.A.; McCue, S.W.; Schugart, R.C.; Dawson, R.A.; McElwain, D.S. Modelling the interaction of keratinocytes and fibroblasts during normal and abnormal wound healing processes. Proc R Soc B: Biol Sci 2012, 279, 3329-3338.
  24. Homey, B.; Steinhoff, M.; Ruzicka, T.; Leung, D.Y. Cytokines and chemokines orchestrate atopic skin inflammation. J Allergy Clin Immunol 2006, 118, 178-189.
  25. Jeremy, A.H.; Holland, D.B.; Roberts, S.G.; Thomson, K.F.; Cunliffe, W.J. Inflammatory events are involved in acne lesion initiation. J Invest Dermatol 2003, 121, 20-27.
  26. Dong, C.; Virtucio, C.; Zemska, O.; Baltazar, G.; Zhou, Y.; Baia, D.; Jones-Iatauro, S.; Sexton, H.; Martin, S.; Dee, J. Treatment of skin inflammation with benzoxaborole phosphodiesterase inhibitors: selectivity, cellular activity, and effect on cytokines associated with skin inflammation and skin architecture changes. J Pharmacol Exp Ther 2016, 358, 413-422.
  27. Aparecida Da Silva, A.; Leal-Junior, E.C.P.; Alves, A.C.A.; Rambo, C.S.; Dos Santos, S.A.; Vieira, R.P.; De Carvalho, P.D.T.C. Wound-healing effects of low-level laser therapy in diabetic rats involve the modulation of MMP-2 and MMP-9 and the redistribution of collagen types I and III. J Cosmet Laser Ther 2013, 15, 210-216.

Please see the additional information in the attachment. 

Round 2

Reviewer 2 Report

Dear Authors

Good Day!

I have checked and read your revised version, I feel there is still matters for us to go the major revision again.

First of all please do not thank me for each comment, I am appreciated of your attention but it is not necessary.

I would like you to intensively pay attention to your abstract and introduction. despite your good work these two section make your paper weaker!

Also try to expand your conclusion to give more insight to your readers about potential of this mushroom. try to be more specific.

Good luck

Author Response

Faculty of Pharmacy

Chiang Mai University,

Chiang Mai 50200, Thailand

December 17th, 2021

Dear Editor-in-Chief, Journal of Fungi,

               We are grateful for your consideration of this manuscript. The valuable reviewer's comments have been extensively analyzed and used to improve the article. As suggested, the manuscript has been revised point by point, shown in a yellow highlight in the reviewer comments' answers. Please find the revised file of the manuscript entitled "High Efficiency In Vitro Wound Healing of Dictyophora indusiata Extracts via Anti-inflammatory and Collagen Stimulating (MMP-2 Inhibition) Mechanisms" (Manuscript ID: jof-1497758) with the track change.

Thank you very much for your kind attention to this matter. I am looking forward to hearing from you.

Sincerely yours,

(Asst. Prof. Dr. Warintorn Ruksiriwanich)
Corresponding Author
E-mail address: warintorn.ruksiri@cmu.ac.th

Faculty of Pharmacy, Chiang Mai University

Tel: +66-96269-5354

Answers to the Comments of the Reviewer

Manuscript ID:                              jof-1497758

Title:                                                 High Efficiency In Vitro Wound Healing of Dictyophora indusiata Extracts via Anti-inflammatory and Collagen Stimulating (MMP-2 Inhibition) Mechanisms

Authors:                                           Nazir et al.

Response to comments:

REVIEWER 2

  1. Extensive editing of English language and style required

Response:

               The authors thoroughly checked the English language and extensively edited it using the Grammarly (premium) program.

  1. I would like you to intensively pay attention to your abstract and introduction. despite your good work these two sections make your paper weaker!

Response:

The abstract has been extremely revised and added the results from the raw data and how each test relates to wound healing process. On page 1 (lines 23-42), the abstract had been changed to “Dictyophora indusiata or Phallus indusiatus is widely used as not only traditional medicine, functional foods, but also, skin care agents. Biological activities of the fruiting body from D. indusiata were widely reported, while the studies on the application of immature bamboo mushroom extracts were limited especially in the wound healing effect. Wound healing process composed of 4 stages including hemostasis, inflammation, proliferation, and remodelling. This study divided the egg stage of bamboo mushroom into 3 parts: peel and green mixture (PGW), core (CW), and whole mushroom (WW). Then, aqueous extracts were investigated for their nucleotide sequencing, biological compound contents, and wound healing effect. The anti-inflammatory determination via the levels of cytokine releasing from macrophages, and the collagen stimulation activity on fibroblasts by matrix metalloproteinase-2 (MMP-2) inhibitory activity were determined to serve for the wound healing process promotion in the stage 2-4 (wound inflammation, proliferation, and remodelling of the skin). All D. indusiata extracts showed promising antioxidant potential, significantly anti-inflammatory activity in the decreasing of the nitric oxide (NO), interleukin-1 (IL-1), interleukin-1 (IL-6), and tumour necrosis factor-α (TNF-α) secretion from macrophage cells (p < 0.05), and the effective collagen stimulation via MMP-2 inhibition. In particular, CW extract containing high content of catechin (68.761 ± 0.010 mg/g extract) which could significantly suppress NO secretion (0.06 ± 0.02 µmol/L) better than the standard anti-inflammatory drug diclofenac (0.12 ± 0.02 µmol/L) and their MMP-2 inhibition (41.33 ± 9.44%) was comparable to L-ascorbic acid (50.65 ± 2.53%). These findings support that CW of D. indusiata could be an essential natural active ingredient for skin wound healing pharmaceutical products.”.

In the introduction, additional information of fungi has been described and the main points of wound healing process have been summarized.  On page 2 (lines 47-96), the introduction had been reconstructed to “Mushroom is rich in nutrients and bioactive compounds. Edible mushroom is regularly being used as food, dietary supplements as well as cosmeceutical products such as anti-ageing, moisturizing, and skin lightening. Since, mushroom polysaccharides and polyphenol contents were involved in antioxidant and immunomodulatory activities [1,2]. Dictyophora indusiata or Phallus indusiatus or bamboo mushroom is a fungus belonging to the family Phallaceae. The bioactive compositions in immature and mature stages of this mushroom are different. The egg or immature mushroom produce viscous mucilage and high phenolic contents[3,4]. The mature stage of this mushroom contains many bioactive compounds, for example, polysaccharides, amino acids, terpenoids, and alkaloids [5,6]. Many studies revealed that the fruiting body of Dictyophora indusiata showed various impressive activities such as anti-obesity [7], and neuroprotective effect for treating Alzheimer’s disease [8]. Furthermore, D. indusiata also has considerable bioactivities with cosmeceutical potentials such as antioxidant, anti-tyrosinase, antimicrobial properties [9-11]. Interestingly, most research papers reported that polysaccharides from the fruiting body of D. indusiata contributed to immune modulation [12-14] and considered as a nutraceutical supplement in Chinese remedy. The superior anti-inflammatory activity of D. indusiata can be applied both via the oral and topical routes. Skin inflammation is a process that happens when the skin is damaged and wounded.

The skin wound repairing process included 4 main approaches, hemostasis, inflammation, proliferation, and tissue remodelling. Firstly, fibrin formation is the crucial step in coagulation cascades to stop blood loss [15]. During this phase, the number of soluble mediators such as platelet-derived growth factor (PDGF), insulin-like growth factor-1 (IGF-1), epidermal growth factor (EGF), fibroblast growth factor (FGF), transforming growth factor-β (TGF-β), and vascular endothelial cell growth factor (VEGF) are released from platelets [16]. Secondly, inflammation phase, macrophages migrate and defend microbes, attract other macrophage cells, and secrete cytokines and protease (elastase and collagenase) which degrade the skin extracellular matrix (ECM) components [17]. Inflammatory cytokines containing interleukin-1 (IL-1) and tumour necrosis factor-α (TNF-α) activated proteases production and apoptosis in fibroblasts. Thirdly, the promotion of cell proliferation and restoration of the matrix, fibroblasts, endothelial cells, and keratinocytes in tissue remodelling are conducted. Growth factors and cytokines are synthesized to promote cell proliferation, develop new capillary formation, and produce new ECM components [18,19]. Subsequently, matrix metalloproteinases (MMPs) will remove injured matrix proteins and then fibroblasts release lysyl oxidase to link collagen at ECM in the scar forming.

However, the most common wound-healing problem is the  hypertrophic scar or keloid resulting from prolonged inflammation [20] and the action of matrix metalloproteinases enzymes [15]. Thus, the suppression of the inflammatory cytokines improves skin structure and barrier functions in the skin wounds [21] and MMP-2 inhibitory potential could restore the balance of collagen production [22] resulting in the successful wound healing process without the hypertrophic scar.

Moreover, most research papers reported various bioactivities of the fruiting body from D. indusiata, while the studies on the application of immature bamboo mushroom extracts as cosmeceutical or pharmaceutical applications were only a few. Therefore, this study aimed to investigate total phenolic, flavonoid, and polysaccharide contents, then further determine the antioxidant capabilities of aqueous extracts from 3 parts of D. indusiata mushroom. In addition, these extracts were assessed for wound healing activity using RAW 264.7 macrophages and hTRT fibroblasts which compared to the standard substances. The results from this study may consider as bioactive sources for pharmaceutical and/or cosmeceutical applications.”.

References:

  1. Taofiq, O.; González-Paramás, A.M.; Martins, A.; Barreiro, M.F.; Ferreira, I.C. Mushrooms extracts and compounds in cosmetics, cosmeceuticals and nutricosmetics—A review. Ind Crops Prod 2016, 90, 38-48.
  2. Wu, Y.; Choi, M.-H.; Li, J.; Yang, H.; Shin, H.-J. Mushroom cosmetics: the present and future. Cosmetics 2016, 3, 22.
  3. Srisuk, N.; Jirasatid, N. Characteristics co-encapsulation of Lactobacillus acidophilus with Dictyophora indusiata. Curr Res Nutr Food Sci 2020, 8, 1013.
  4. Burapapadh, K.; Changsan, N.; Sinsuebpol, C.; Saokham, P. An evaluation of Dictyophora indusiata mucilage as a binder in tablet formulations. Key Eng Mater 2021, 901, 22-27.
  5. Habtemariam, S. The chemistry, pharmacology and therapeutic potential of the edible mushroom Dictyophora indusiata (Vent ex. Pers.) Fischer (Synn. Phallus indusiatus). Biomedicines 2019, 7, 98.
  6. Wang, J.; Wen, X.; Zhang, Y.; Zou, P.; Cheng, L.; Gan, R.; Li, X.; Liu, D.; Geng, F. Quantitative proteomic and metabolomic analysis of Dictyophora indusiata fruiting bodies during post-harvest morphological development. Food Chem 2021, 339, 127884.
  7. Wang, W.; Song, X.; Zhang, J.; Li, H.; Liu, M.; Gao, Z.; Wang, X.; Jia, L. Antioxidation, hepatic-and renal-protection of water-extractable polysaccharides by Dictyophora indusiata on obese mice. Int J Biol Macromol 2019, 134, 290-301.
  8. Talebi, M.; Kakouri, E.; Talebi, M.; Tarantilis, P.A.; Farkhondeh, T.; İlgün, S.; Pourbagher-Shahri, A.M.; Samarghandian, S. Nutraceuticals-based therapeutic approach: recent advances to combat pathogenesis of Alzheimer’s disease. Expert Rev Neurother 2021, 21, 625-642.
  9. Oyetayo, V.; Dong, C.-H.; Yao, Y.-J. Antioxidant and antimicrobial properties of aqueous extract from Dictyophora indusiata. Open Mycol J 2009, 3.
  10. Sharma, V.K.; Choi, J.; Sharma, N.; Choi, M.; Seo, S.Y. In vitro anti‐tyrosinase activity of 5‐(hydroxymethyl)‐2‐furfural isolated from Dictyophora indusiata. Phytother Res: An International Journal Devoted to Pharmacological and Toxicological Evaluation of Natural Product Derivatives 2004, 18, 841-844.
  11. Hua, Y.; Yang, B.; Tang, J.; Ma, Z.; Gao, Q.; Zhao, M. Structural analysis of water-soluble polysaccharides in the fruiting body of Dictyophora indusiata and their in vivo antioxidant activities. Carbohydr Polym 2012, 87, 343-347.
  12. Liao, W.; Luo, Z.; Liu, D.; Ning, Z.; Yang, J.; Ren, J. Structure characterization of a novel polysaccharide from Dictyophora indusiata and its macrophage immunomodulatory activities. J Agric Food Chem 2015, 63, 535-544.
  13. Deng, C.; Shang, J.; Fu, H.; Chen, J.; Liu, H.; Chen, J. Mechanism of the immunostimulatory activity by a polysaccharide from Dictyophora indusiata. Int J Biol Macromol 2016, 91, 752-759.
  14. Wang, Y.; Lai, L.; Teng, L.; Li, Y.; Cheng, J.; Chen, J.; Deng, C. Mechanism of the anti-inflammatory activity by a polysaccharide from Dictyophora indusiata in lipopolysaccharide-stimulated macrophages. Int J Biol Macromol 2019, 126, 1158-1166.
  15. Shih, B.; Garside, E.; McGrouther, D.A.; Bayat, A. Molecular dissection of abnormal wound healing processes resulting in keloid disease. Wound Repair Regen 2010, 18, 139-153.
  16. Schultz, G.S.; Sibbald, R.G.; Falanga, V.; Ayello, E.A.; Dowsett, C.; Harding, K.; Romanelli, M.; Stacey, M.C.; Teot, L.; Vanscheidt, W. Wound bed preparation: a systematic approach to wound management. Wound Repair Regen 2003, 11, S1-S28.
  17. Enoch, S.; Leaper, D.J. Basic science of wound healing. Surgery (Oxford) 2005, 23, 37-42.
  18. Buranasukhon, W.; Athikomkulchai, S.; Tadtong, S.; Chittasupho, C. Wound healing activity of Pluchea indica leaf extract in oral mucosal cell line and oral spray formulation containing nanoparticles of the extract. Pharm Biol 2017, 55, 1767-1774.
  19. Ruksiriwanich, W.; Khantham, C.; Linsaenkart, P.; Jantrawut, P.; Rajchasom, S. Optimization of placenta extraction for wound healing activity. Chiang Mai J Sci 2019, 46, 946-959.
  20. Menon, S.N.; Flegg, J.A.; McCue, S.W.; Schugart, R.C.; Dawson, R.A.; McElwain, D.S. Modelling the interaction of keratinocytes and fibroblasts during normal and abnormal wound healing processes. Proc R Soc B: Biol Sci 2012, 279, 3329-3338.
  21. Dong, C.; Virtucio, C.; Zemska, O.; Baltazar, G.; Zhou, Y.; Baia, D.; Jones-Iatauro, S.; Sexton, H.; Martin, S.; Dee, J. Treatment of skin inflammation with benzoxaborole phosphodiesterase inhibitors: selectivity, cellular activity, and effect on cytokines associated with skin inflammation and skin architecture changes. J Pharmacol Exp Ther 2016, 358, 413-422.
  22. Aparecida Da Silva, A.; Leal-Junior, E.C.P.; Alves, A.C.A.; Rambo, C.S.; Dos Santos, S.A.; Vieira, R.P.; De Carvalho, P.D.T.C. Wound-healing effects of low-level laser therapy in diabetic rats involve the modulation of MMP-2 and MMP-9 and the redistribution of collagen types I and III. J Cosmet Laser Ther 2013, 15, 210-216.

  1. Also try to expand your conclusion to give more insight to your readers about potential of this mushroom. try to be more specific.

Response:

The conclusion has been explained more details about the abilities of this mushroom. On page 14 (lines 483-500), the sentences have been revised to “Since there is no publication of the in vitro wound healing effect of immature (egg) bamboo mushroom extracts, the anti-inflammatory and collagen stimulating (MMP-2 inhibition) activities for the wound healing were done in this study. The aqueous extracts from the immature stage of core D. indusiata mushroom showed the considerable wound healing effect not only with the highest antioxidant activity, the comparable anti-inflammation via the reduction of cytokines (NO, IL-1, IL-6, and TNF-α) secretion from LPS-induced macrophage cells to the standard diclofenac, but also the promising MMP-2 inhibition through the determination of gelatinolytic activity on fibroblasts cells which were responsible for inflammation, cell proliferation, and tissue remodelling stages in the wound healing process, respectively. Over-production of inflammatory cytokines and MMP-2 lead to wound healing impairment, chronic skin inflammatory diseases, and scar formation. Therefore, CW which provided the high content of catechin, other polyphenolic compounds, and the effective anti-inflammation and MMP-2 inhibition activity could have the promising in vivo wound healing effect could be a candidate to develop as pharmaceutical and/or cosmeceutical wound healing active ingredients. Hence, the in vivo study of CW extract should be further investigated in animal and volunteers with wound injury for the honorable result. The animal and human ethical approval are currently on the application process.”.

We are very much appreciated your suggestion. All comments we received have been taken to improve the quality of the article.

Sincerely,

Nazir et al.

This manuscript is a resubmission of an earlier submission. The following is a list of the peer review reports and author responses from that submission.

Round 1

Reviewer 1 Report

The manuscript High Efficiency In Vitro Wound Healing of Dictyophora indusiata Extracts via Anti-inflammatory and Collagen Stimulating (MMP-2 Inhibition) Mechanisms is very well conceptualized with well-defined research objectives. Methods are detailed and all figures and tables in the paper properly represent the results. Accept with minor changes.

Reviewer 2 Report

This is an interesting research with promising results. However, many problems were detected that prevent an unquestionable publication of the article, methodology, results, and conclusions. The scientific name of this fungus is actually Phallus indusiatus and the name Dictyophora indusiata is a synonym. This update is necessary. There is a lack of an identification of the fungus by a taxonomist or better by molecular techniques to ensure that the fungus cited is the same one. A deposit of the strain in an international bank would be convenient and some of them do this kind of identification.
In general the English writing contains parts that do not make sense and should be revised in general. i.e. "Then, the extracts were determined their bioactive components and antioxidant activities." Other parts should be translation errors or are conceptual errors. i.e. "The functions of skin are to protect our organ from environment like microorganism and allergen, maintain internal environment such as hormones, water balance, and temperature [1]."
The scientific name of the fungus was often written without the italics in violation of the taxonomic rule of scientific writing of microorganisms. At another point the text implies that the fungus is a plant or there is a comparison with a plant from the literature, however the English writing lacks the clarity or the content is inadequate. i.e. "However, the PGW also lower activity than mechanical plant extract for DPPH, ABTS, and FRAP [27,28]."
Description of the parts of the basidiocarp that were used for the extracts is incomplete or confusing, not allowing clarity and understanding for analysis. Description of the mushroom cultivation (or if it was collected in the wild) such as substrate formulation and cultivation conditions, which directly affects the interpretation of the results and discussion, was not presented.
In the results there is a lack of controls for comparison and the only control used is inadequate. i. e. Table 3. Figure 1 has interesting data, but difficult to visualize or confusing and therefore a new presentation should be proposed. Interpretation of the results and inference based on statistics is biased and needs to be reformulated for credibility of the results found.
The discussion was not properly developed.
Despite having positive results for conclusion, the writing is inadequate and with inferences beyond what the results indicate.